# Optimization of 'on farm' hydropriming conditions in wheat: Soaking time and water volume have interactive effects on seed performance

**Hemender Tanwar**[1]\*, **Virender Singh Mor**[1], **Sushma Sharma**[1], **Mujahid Khan**[2], **Axay Bhuker**[1], **Vikram Singh**[3], **Jitender Yadav**[1], **Sonali Sangwan**[4], **Jogender Singh**[5], **Shikha Yashveer**[4], **Kuldeep Singh**[6]

**1** Department of Seed Science and Technology, Chaudhary Charan Singh Haryana Agricultural University, Hisar, India, **2** Agricultural Research Station (S.K.N. Agriculture University, Jobner), Fatehpur Shekhawati, Sikar, Rajasthan, India, **3** Department of Genetics and Plant Breeding, Chaudhary Charan Singh Haryana Agricultural University, Hisar, India, **4** Department of Molecular Biology, Biotechnology and Bioinformatics, Chaudhary Charan Singh Haryana Agricultural University, Hisar, India, **5** Krishi Vigyan Kendra, Fatehabad, Chaudhary Charan Singh Haryana Agricultural University, Hisar, India, **6** Krishi Vigyan Kendra, Sonepat, Chaudhary Charan Singh Haryana Agricultural University, Hisar, India

\* hemender@hau.ac.in

**Data Availability Statement:** All relevant data are within the manuscript and its Supporting information files.

## Abstract

Seed priming is a simple and cost effective method to obtain a better plant stand under diverse environmental conditions. The current study was designed to determine the optimal priming duration and water volume for wheat seed. For this experiment, three wheat genotypes with distinct genetic and adaptive backgrounds were chosen. Seeds of each genotype were hydroprimed for 7 durations, i.e. 1, 2, 4, 8, 12, 16, and 20 hours, in three different water volumes, i.e. half, equal, and double volume with respect to seed weight and then surface dried for 1 hour. The control was unprimed (dry) seed. The germination characteristics and seedling vigour potential of hydroprimed seeds were evaluated in the lab by recording several parameters such as germination percentage and speed, seedling growth, and vigour indices at two different temperature levels. The results showed that optimal duration for hydropriming of wheat seed is 12 hours with an equal volume with respect to original seed weight, closely followed by 8 hours with double volume. Reduction in seed performance was observed at 16 and 20 hours priming particularly at double volume treatment. Effect of temperature on seed germination showed improvement in seedling vigour at 25°C when compared to 20°C, although effect on germination percentage was non-significant. Volume of water and priming duration showed significant interactive effects demonstrating that a higher volume can give equivalent results at a shorter duration and vice versa. Another experiment was also conducted to compare the on-farm priming (surface dried seed) with conventional priming (seed re-dried to original moisture) taking 3 potential durations i.e. 8, 12 and 16 hours. Results revealed that both priming methods were statistically at par in terms of germination percentage, while, surface drying resulted in better seedling vigour and speed of germination.

**Funding:** The author(s) received no specific funding for this work.

## Introduction

Enhancement of seed performance by conditioning them in simple water (seed hydropriming) has been known for many years [1]. Seed priming accelerates pre-germination metabolism, resulting in faster seedling emergence and a more uniform plant stand in the field [2, 3]. Primed seeds quickly absorb water, accompanied by ion and solute leakage, which stimulates repair of structural and oxidative damage caused by desiccation, and membrane integrity is restored, resulting in rejuvenation of respiration and metabolism [4]. Seeds that have been well-primed germinate faster and more evenly than unprimed seeds, and priming also confers stress tolerance at an early stage of plant growth.

Hydropriming is the most basic, cost-effective, and ecologically safe technique for improving seed performance in the field, enhancing plant stand, and crop output under both stressed and normal conditions [5]. Classical method of seed hydropriming involves seed soaking in pure water followed by re-drying to the initial moisture level before to sowing. Changes caused by hydropriming can be classified into three distinct phases [6]. Due to the water potential difference in the system, the seed absorbs water quickly in Phase I (seed having lower water potential). The key events occurring during this phase include DNA and mitochondrial repairs, as well as the synthesis of new protein molecules from existing mRNA. In phase II, seed metabolism gets activated, mitochondrial synthesis begins, and protein synthesis is accelerated via translation of newly generated mRNAs. Phase III is characterized by rapid cell division and elongation, which aids in tissue growth and leads to germination, i.e. radicle protrusion. However, priming only allows seed hydration throughout phase I and before the completion of phase II, where germination remains a reversible process and stops just short of radicle protrusion [7].

There are some limitations in adoption of conventional hydropriming at farmers' field. Most farmers in poor and developing countries have little resources and technical knowledge. Cereal crops such as wheat, barley, and paddy necessitate a larger quantity of seed. Soaking and then drying such seeds to their original moisture content before sowing complicates the process for farmers. Therefore, 'On-farm' hydropriming becomes a popular approach among farmers because it is simple to implement in real farm situations. It is a type of hydropriming in which seed is soaked in water, dried for a brief period, and then sown. The period of treatment must not exceed the "safe limit" (the maximum time of priming without risk of seed or seedling damage due to premature germination) [8]. The positive impact of this method on crop emergence and yield was confirmed by Harris et al. [9]. On-farm priming is particularly beneficial for resource-constrained farmers in marginal tropical environments [10]. However, a drawback of this procedure is that seeds would absorb water in unregulated manners as farmers use non-standardized method (generally soaking the seed overnight). As a result, seeds have unrestricted access to water, and the tissue affinity of the seed to water is the only factor determining the rate of seed water intake [11].

Practically, the recommendation of standardized soaking period is important for seed priming of wheat because (a) farmer usually soak the seed in evening hours and take them out in morning (1–2 hours after sunrise), so as the nights are longer during sowing season, there is a risk of overpriming. (b) The resource poor farmers have limited availability of sowing implements. There are high chances that the sowing is not done on time, hence the oversoaking of seed is likely to happen. This approach may cause non-uniform seed hydration, resulting in unsynchronized metabolic activation in seeds and non-homogeneous emergence [12]. If the seed's water absorption during hydropriming is not carefully maintained to a safe limit, it can injure the seed and cause it to lose vigour.

Furthermore, the water absorption capacity and response to hydropriming of different crop species/genotypes vary in nature. These constraints emphasize the significance of standardizing the specific conditions of this procedure, such as treatment duration, temperature, and water volume, in order to achieve the optimal seed hydration. Therefore, the current study was aimed to determine the ideal soaking duration and water volume for hydropriming of wheat seed which can be implemented at the farm level. Comparison of on-farm priming was also done with the conventional method by taking potential soaking durations to further validate the results.

## Materials and methods

### Seed material and priming treatments

The seed of three Indian bread wheat genotypes (WH 1105, WH 1124 and KRL 213) was obtained from Wheat and Barley Section, Department of Genetics and Plant Breeding, Chaudhary Charan Singh Haryana Agricultural University, Hisar. The genotypes were selected based on their diverse genetic backgrounds and adaptation to different growing environments (Table 1). They represent a standard sample from seed supply chain of wheat in the area. Seeds were stored in ambient conditions prior the experiment.

### Experiment 1

The priming treatments were applied to all combinations and seeds were soaked in tap water in 100 ml glass beakers, at 25°C in dark. The beakers were kept on a mechanical shaker to prime the seeds uniformly. After treatment, seeds were allowed to air-dry on a paper towel for an hour (surface drying) to avoid clumping. In all the cases, non-primed dry seed was used as control. Twenty gram seed of each genotype was soaked in 10 (Half volume (w/v), 20 (Equal volume (w/v) and 40 ml (Double volume (w/v) water for either 1, 2, 4, 8, 12, 16, or 20 h in triplicate for each soaking time. Three samples of unsoaked seeds (10 g each) were oven-dried at 103°C for 17 h to determine initial moisture content (Mc) [16]. All the seed samples were weighed before soaking. Seeds were weighed again after soaking and a brief drying (1 hour) for determination of their final moisture content (Mc) after priming, which was calculated as follows:

$$Mc = \left(\frac{M_2 - M_3}{M_2 - M_1}\right) \times 100$$

Where, $M_1$ is weight of empty container, $M_2$ is weight of container with seed before drying in the oven and $M_3$ is weight of container with seed after drying the oven.

Water Exposure Index (WEI) during different combinations of hydropriming conditions was calculated by using following formula:

$$WEI = \sqrt{wv} \times sd$$

**Table 1. Details of genotypes used in the study.**

| Sr. No. | Name of the genotype | Year of release | Parentage | Recommended ecology |
|---------|---------------------|-----------------|-----------|---------------------|
| 1. | WH 1105 | 2013 | MILAN/S87230//BABAX | Normal sown, irrigated [13] |
| 2. | WH 1124 | 2014 | MUNIA/CHTO//AMSEL | Late sown, irrigated [14] |
| 3. | KRL 213 | 2010 | CNDO/R143/ENTE/MEXI-1-1/3/*Ae. squarrosa* (TAUS)/4/Weaver/5/2*Kauz | Salt tolerant [15] |

Where, *wv* is water volume denoting fixed values either 0.5 (half volume), 1 (equal volume) or 2 (double volume) and *sd* is the soaking duration in minutes.

## Experiment 2

Another experiment was designed to compare the on farm priming with conventional priming in which seeds were dried back to their original moisture content. Three priming durations viz. 8, 12 and 16 hours were selected on the basis of experiment 1. Priming treatments were given in the similar fashion as experiment 1 and twenty gram seed of each genotype was soaked in 10, 20 and 40 ml water for 8, 12, and 16 hours in triplicates for each soaking duration. After treatment, one set of seeds was surface dried for 1 hour and another set was air-dried for 24 hours to dry them to their original moisture content ($\approx$11%).

## Experimentation

For both experiments, standard germination percentage and seedling related traits were recorded by using 'Between Paper' method [16] for germination testing. One hundred healthy unbroken seeds of each genotype were taken and placed equidistantly between two sufficiently moistened towel papers. These towel papers were then rolled and covered with a layer of wax paper to avoid the moisture loss and kept on steel racks in growth chambers maintained at 20˚C and 25˚C in dark conditions for 8 days.

For assessing the speed of germination, 'Top of the Paper' method was used. Fifty seeds were planted on moistened filter paper kept in plastic petri plates. These petri plates were then kept in the germinators under the same experimental conditions for 8 days. The relative humidity during the course of experiments was maintained at 90±2%. The experiments were laid out in a completely randomized design in a factorial arrangement and replicated thrice.

## Observations recorded

The final count of germination was taken on $8^{th}$ day and normal seedlings were considered for percent germination [16] and values were expressed in percentage. The newly emerged radicals of germinated seeds were counted on a daily basis. Speed of germination was calculated based on the following formula given by Maguire [17]:

$$\text{Speed of germination} = \frac{X1}{Y1} + \frac{X2 - X1}{Y2} + \cdots + \frac{Xn - Xn - 1}{Yn}$$

Where,
$X_1$, $X_2$ and $X_n$ = number of seeds germinated on the first, second and $n^{th}$ day, respectively
$Y_1$, $Y_2$ and $Y_n$ = number of days from sowing to first, second and $n^{th}$ count, respectively

Thirty seedlings were randomly selected from the 'Between Paper' samples and their shoot and root lengths were measured at 8 DAS. Average of the 30 seedlings was taken for the final calculation. Fresh weight of seedlings from each replicate was also recorded immediately. For the estimation of dry weight, the seedlings whose fresh weight was recorded were dried in a hot air oven for 24 hours at 80±1˚C. The dried seedlings of each replication were weighed and dry weight of single seedling was calculated by taking the average for each and expressed in milligrams. The seedling vigour index-I and vigour index-II were calculated by the formulae

suggested by Abdul-Baki and Anderson [18] and expressed as a whole number.

$$\text{Seedling Vigour Index-I} = \text{Standard germination}\,(\%) \times \text{Average seedling length}\,(cm)$$

$$\text{Seedling Vigour Index-II} = \text{Standard germination}\,(\%) \times \text{Average seedling dry weight}\,(mg)$$

Hydropriming Optimization Score (HPOS) was determined using Standard Germination (SG) and Germination Speed (GS), using the following formula:

$$HPOS = \frac{2 \times SG \times GS}{SG + GS}$$

This formula was developed after modifying the basic formula described by Liu et al. [19] for Lucerne seeds.

## Statistical analysis

The data from different treatment combinations are presented as the mean value with standard error (error bars) of three replicates in the tables or graphical form. All the data were analyzed in Completely Randomized Design (CRD) using STAR 5.1: Statistical Tool for Agricultural Research of International Rice Research Institute (IRRI). Four-way ANOVA was used to detect the effect of genotype, water volume, soaking duration and temperature on different seed germination and seedling vigour parameters. Least significant difference (LSD) test was used at 0.05 probability levels to check the difference between different treatments. Only significant interaction effects are presented with the help of tables and graphs. Correlation analysis was also done to investigate the relation of seed moisture content and water exposure index with germination and seedling vigour traits of primed seeds.

## Results

### Experiment 1

**Imbibition pattern.**   Water imbibition at different water volumes of soaking and duration demonstrated that half volume with respect to seed weight is insufficient to achieve maximum absorption. While imbibition in seeds soaked in equal and double volumes resulted in the same amount of water absorbed, the rate of absorption differed under each volume treatment. The difference between these two volume treatments was significant up to 8 hours of soaking duration, but it became inconsequential at 12, 16, and 20 hours. Typically, the wheat seed absorbed roughly 50 percent water with regard to its weight upto 8 hours of soaking and the general peak in water absorption was noted at 16 hours, following which there was a very small augmentation at 20 hours (Fig 1).

Moisture content following hydropriming also exhibited a similar pattern (Fig 2). The moisture content of dry seed (control) was approximately 10%. Moisture content increased sharply during the first hour of priming in all three volume treatments. Their magnitudes of increase, however, were different, and half volume treatment resulted in a lower increase. Moisture content increased to a maximum of 16 hours after priming and then stabilized. Moisture was somewhat comparable between equal and double volume treatments with double volume having a slightly higher value and peaking at around 40% at 16 hours. Whereas, half volume demonstrated significantly lower moisture contents across all durations, reaching a maximum of 35%.

**Germination characteristics.**   Germination tests were performed at two temperature levels on each cultivar to determine the most promising duration of soaking at three water

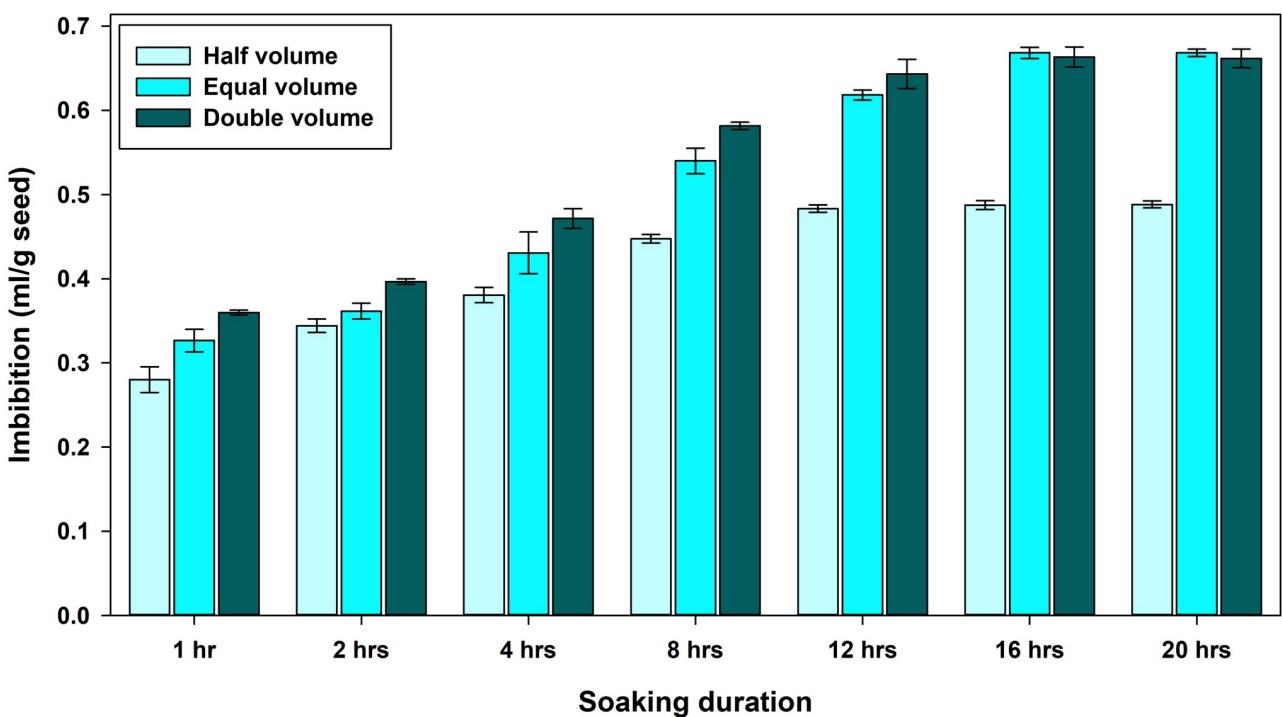

**Fig 1. Imbibition pattern during hydropriming under different set of soaking durations and water volumes.**

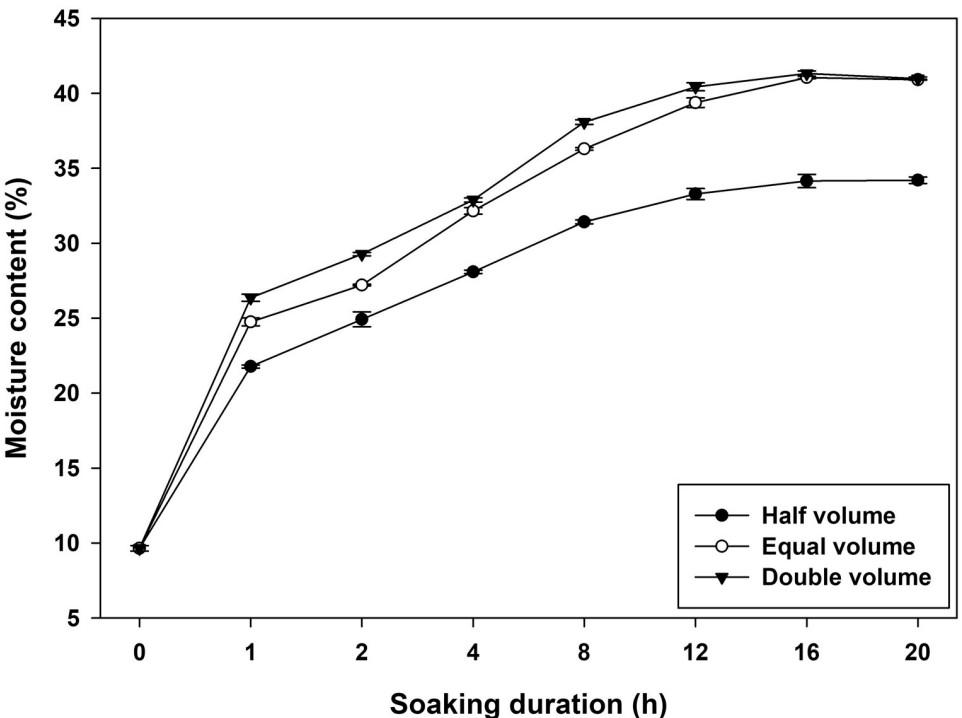

**Fig 2. Seed moisture content as influenced by soaking time and water volume during hydropriming of wheat.**

**Table 2. Analysis of variance (F-value) for effects of genotype, water volume, soaking duration and temperature on germination characteristics, seedling growth and vigour indices in wheat seeds.**

| Source of variation | DF | Standard germination | Germination speed | Shoot length | Root length | Seedling length | Seedling fresh weight | Seedling dry weight | Seedling vigour index-I | Seedling vigour index-II |
|---|---|---|---|---|---|---|---|---|---|---|
| Genotype | 2 | 41.75** | 52.83** | 178.12** | 239.78** | 54.84** | 967.50** | 659.19** | 43.90** | 533.24** |
| Volume | 2 | 1.19 | 22.12** | 8.26** | 5.63** | 10.06** | 3.13* | 1.78 | 7.31 | 0.71 |
| Soaking duration | 7 | 267.38** | 829.52** | 130.99** | 71.60** | 140.86** | 102.55** | 58.91** | 275.32** | 193.93** |
| Temperature | 1 | 2.27 | 418.10** | 3930.81** | 1315.70** | 3280.12** | 256.68** | 1239.75** | 1667.12** | 921.21** |
| Genotype × Water volume | 4 | 1.04 | 2.33 | 1.93 | 2.75* | 3.23* | 3.42** | 5.99** | 1.21 | 3.30* |
| Genotype × Soaking duration | 14 | 3.50** | 0.34 | 9.72** | 5.02** | 9.06** | 7.41** | 2.07* | 3.32** | 1.31 |
| Genotype × Temperature | 2 | 0.73 | 0.04 | 24.86** | 32.07** | 44.35** | 34.79** | 65.65** | 19.33** | 50.85** |
| Water volume × Soaking duration | 14 | 4.58** | 39.12** | 6.38** | 4.72** | 7.89** | 3.43** | 6.97** | 6.37** | 6.34** |
| Water volume × Temperature | 2 | 0.33 | 0.71 | 0.68 | 1.81 | 1.36 | 1.22 | 0.62 | 0.67 | 0.12 |
| Soaking duration × Temperature | 7 | 0.24 | 0.71 | 6.09** | 8.51** | 9.12** | 1.29 | 1.09 | 5.80** | 1.74 |
| Genotype × Water volume × Soaking duration | 28 | 0.29 | 0.32 | 1.08 | 0.64 | 0.57 | 0.62 | 0.44 | 0.44 | 0.46 |
| Genotype × Water volume × Temperature | 4 | 0.39 | 0.11 | 1.36 | 2.62* | 1.42 | 0.52 | 1.46 | 1.51 | 1.44 |
| Genotype × Soaking duration × Temperature | 14 | 0.32 | 0.17 | 7.39** | 5.83** | 5.31** | 1.00 | 0.63 | 3.26** | 0.47 |
| Water volume × Soaking duration × Temperature | 14 | 0.11 | 0.09 | 0.27 | 0.14 | 0.21 | 0.30 | 0.28 | 0.12 | 0.20 |
| Genotype × Water volume × Soaking duration × Temperature | 28 | 0.33 | 0.06 | 0.55 | 0.43 | 0.31 | 0.25 | 0.32 | 0.20 | 0.42 |

**Significant at $p = 0.01$,

*Significant at $p = 0.05$

volumes of priming. Germination percentage was significantly influenced by varied soaking durations ($p<0.01$) (Table 2), and it gradually rose with time up to 12 hours. However, the value was statistically equivalent to that obtained at 8 hours. Germination decreased at the duration more than 12 hours, with the lowest recorded at 20 hours (even lower than unprimed dry seed) (Table 3). In terms of germination speed, the impacts were comparable, with seeds primed for 12 hours germinating faster. Although the speed was slower at 16 and 20 hours, the seeds germinated faster than unprimed seeds (Table 3).

Different water volume treatments had no effect on final germination percentage ($p = 0.82$) (Table 2). However, their influence was significant in terms of germination speed ($p<0.01$), with equal volume of water being the fastest followed by double volume treatment, and seeds primed with half volume germinated at a slower rate. For both these traits, the interactive effect of volume and duration was also significant (Table 2). Highest number of seeds germinated after 12 hours of hydropriming with half volume of water, followed by 12 hours of equal volume and 8 hours of double volume. When primed in equal and half amounts of water, the seeds germinated faster at 12 hours, however at double volume, 8 hours was observed to be the optimum (Fig 3).

Genotypic differences were also found to be significant ($p<0.01$) (Table 2), with WH 1105 outperforming WH 1124 and KRL 213 in terms of final germination percentage. However, WH 1124 outperformed the other two genotypes in terms of germination speed. The genotype KRL 213 performed poorly for both germination percentage and speed (Table 3). Effect of

**Table 3. Main effects of genotype, water volume, soaking duration and temperature on standard germination (%) and germination speed of hydroprimed wheat seed.**

| Main effects | Levels | Standard germination (%) | Germination speed |
|---|---|---|---|
| Genotype, gp | WH 1105 | 94.46 a | 63.38 b |
| | WH 1124 | 93.70 b | 64.93 a |
| | KRL 213 | 92.07 c | 61.45 c |
| Water volume, wv | Half | 93.65 a | 62.21 c |
| | Equal | 93.27 a | 64.46 a |
| | Double | 93.32 a | 63.08 b |
| Soaking duration, sd | 0 h (Control) | 92.89 e | 44.04 g |
| | 1 h | 94.30 d | 54.13 f |
| | 2 h | 95.26 cd | 61.13 d |
| | 4 h | 95.82 bc | 69.11 c |
| | 8 h | 97.15 ab | 74.55 b |
| | 12 h | 97.47 a | 77.62 a |
| | 16 h | 92.70 e | 68.58 c |
| | 20 h | 81.71 f | 56.84 e |
| Temperature, Tm | 20°C | 93.57 a | 60.41 b |
| | 25°C | 93.25 a | 66.09 a |
| $LSD_{gp}$ | | 0.63 | 0.654 |
| $LSD_{wv}$ | | NS | 0.654 |
| $LSD_{sd}$ | | 1.332 | 1.383 |
| $LSD_{Tm}$ | | NS | 0.446 |
| d.f. | | 288 | 288 |

Values with different letters within a column (for each main effect) differ significantly from each other (P < 0.05).

LSD, least significant differences between the treatments; d.f., degrees of freedom for the residual term; NS, Non-significant

temperature was non-significant for germination percentage (p = 0.133) while it was significant (p<0.01) for germination speed. Main effect of temperature indicated a higher pace of germination at 25°C in comparison to 20°C level (Table 3).

The interaction between genotype and soaking duration was significant (p<0.01) for standard germination but it was non-significant for germination speed (p = 0.987). This

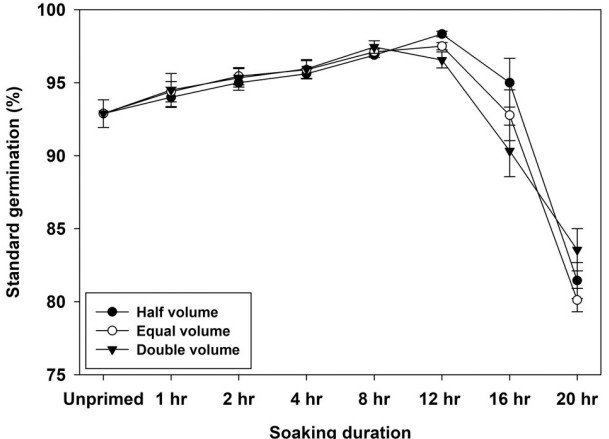
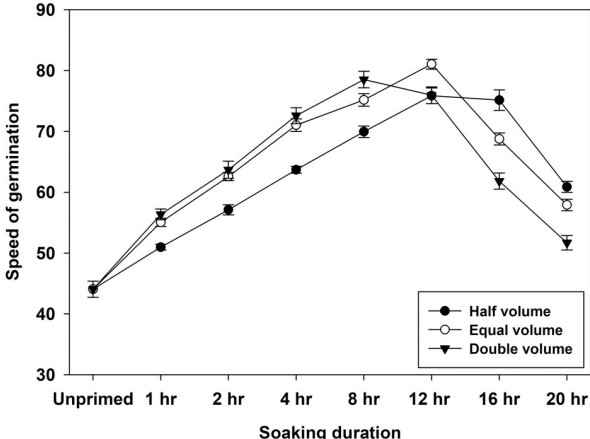

**Fig 3. Interactive effects of water volume and soaking duration of hydropriming on germination characteristics of wheat seed.**

**Table 4. Main effects of genotype, water volume, soaking duration and temperature on seedling growth characteristics in hydroprimed wheat seed.**

| Main effects | Levels | Shoot length (cm) | Root length (cm) | Seedling length (cm) |
|---|---|---|---|---|
| Genotype, gp | WH 1105 | 9.86 b | 19.40 b | 29.26 c |
| | WH 1124 | 9.29 c | 21.06 a | 30.35 a |
| | KRL 213 | 10.12 a | 19.39 b | 29.51 b |
| Water volume, wv | Half | 9.83 a | 20.09 a | 29.91 a |
| | Equal | 9.79 a | 19.95 ab | 29.72 a |
| | Double | 9.65 b | 19.81 b | 29.45 b |
| Soaking duration, sd | 0 h (Control) | 8.68 g | 18.91 e | 27.48 g |
| | 1 h | 9.54 e | 19.55 d | 29.09 e |
| | 2 h | 9.74 de | 19.84 cd | 29.58 de |
| | 4 h | 9.90 cd | 20.33 b | 30.22 c |
| | 8 h | 10.29 b | 20.84 a | 31.13 b |
| | 12 h | 10.58 a | 21.10 a | 31.68 a |
| | 16 h | 10.01 c | 20.01 bc | 30.02 cd |
| | 20 h | 9.31 f | 19.02 e | 28.32 f |
| Temperature, Tm | 20°C | 8.61 b | 18.72 b | 27.30 b |
| | 25°C | 10.91 a | 21.18 a | 32.08 a |
| $LSD_{gp}$ | | 0.105 | 0.199 | 0.241 |
| $LSD_{wv}$ | | 0.105 | 0.199 | 0.241 |
| $LSD_{sd}$ | | 0.223 | 0.421 | 0.511 |
| $LSD_{Tm}$ | | 0.072 | 0.136 | 0.165 |
| d.f. | | 288 | 288 | 288 |

Values with different letters within a column (for each main effect) differ significantly from each other (P < 0.05).

LSD, least significant differences between the treatments; d.f., degrees of freedom for the residual term.

interaction indicated that for shorter durations of priming (1,2 and 4 hours), genotypes WH 1105 and WH 1124 showed statistically similar germination percentage which was higher than KRL 213. While, at priming durations of 8 and 12 hours, these three genotypes were statistically at par with each other. However, at longer priming durations (16 and 20 hours), germination declined in all the three genotypes showing statistically different values with highest in WH 1105 followed by WH 1124 and KRL 213. Similarly, the interaction between water volume and soaking duration was significant (p<0.01) for both the traits i.e. germination percentage and speed. This interaction showed that maximum values were recorded in 12 hours duration at half and equal volume while at double volume, highest germination percentage and speed were found at 8 hours priming.

**Seedling growth and biomass.** On the day of the final count of standard germination test, seedling growth metrics such as shoot, root, and seedling length were recorded. The volume treatments had a significant (p<0.01) effect on shoot, root and seedling length (Table 2). Half and equal volume effects were statistically similar and higher than double volume, indicating that soaking in half and equal volume is better for seedling growth than soaking in double volume (Table 4). The genotypic variations influenced seedling growth metrics significantly (p<0.01). Shoots were longer in genotypes WH 1105 and KRL 213 than in WH 1124. On the other hand, WH 1124 produced longer roots than the other two genotypes (Table 4). In terms of priming duration, the maximum values of shoot and seedling length were recorded after 12 hours of hydropriming, followed by 8 hours. Similarly, in case of root length, 12 hours priming resulted in highest value numerically but it was statistically at par with 8 hours duration. Effect of temperature was also significant (p<0.01) for all the seedling growth parameters.

Significantly higher shoot, root and seedling length was recorded at 25˚C as compared to 20˚C (Table 4).

The interaction between duration and amount of water was found significant (p<0.01). This interaction effect indicated that seedling development was higher at 12 hours with half volume treatment, closely followed by similar duration with equal volume (Fig 4). At double volume, however, numerically highest values of seedling growth parameters were recorded at 12 hours but it was statistically similar with 8 hours duration. All priming treatments significantly increased shoot length compared to the control (unprimed seed) at both temperature levels, but 20 hours duration exhibited a lower value than the control at 25˚C.

The interaction between genotype and water volume was also found significant for root (p = 0.028) and seedling length (p = 0.013). This interaction revealed that the genotypes WH 1105 and KRL 213 produced longer roots and seedlings at half volume treatment and lowest at double volume. While, WH 1124 had no significant difference among the volume treatments for these traits (S1 Table). For the first order interaction of genotype and soaking duration, the genotypes WH 1105 outperformed other two genotypes at all the levels of soaking duration for shoot length. While, the genotype WH 1124 outperformed other genotypes at all the levels of soaking duration for root length and seedling length. For seedling growth, highest performance was obtained at 12 hours soaking duration for all the three genotypes (S2 Table). The interaction between genotype and temperature was also significant (p<0.01) which indicated that at 20˚C, shoot length was maximum in WH 1105 while at 25˚C, it was maximum in KRL 213. Similarly, for root and seedling length, WH 1105 showed second rank at 20˚C but it performed worst at 25˚C among the three genotypes (S3 Table). For the first order interaction between soaking duration and temperature, significant (p<0.01) difference between control (unprimed) and different priming duration was observed at 20˚C. However, at 25˚C, unprimed seeds showed statistically similar values with 1 hour (for shoot and root lengths), 16 hours (for root length) and 20 hours durations (for root and seedling lengths) (S7 Table).

For the second order interaction among genotype, soaking duration and temperature; the gain in shoot length, root length, seedling length and vigour index-I was found at 25˚C with respect to 20˚C for all the three genotypes at all the soaking duration levels, whereas, the highest shoot length, root length, seedling length and vigour index-I were attained at 12 hours soaking duration for all the genotypes at both temperature levels (S6 Table). The effect of second order interaction among genotype, water volume and temperature was found significant (p = 0.035) only for the root length. This interaction indicated that the genotypes WH 1105 and KRL 213 produced longer roots when primed in half volume at 20˚C, while, WH 1124 produced maximum root length at double volume treatment. At 25˚C, no significant difference was observed for root length of WH 1105 and KRL 213 under different volume treatments. However, WH 1124 produced significantly longer roots at equal volume treatment (S6 Table).

The amount of biomass accumulated in seedlings was calculated by recording their fresh and dried weight at 8 DAS (days after sowing). Water volume showed a significant effect (p = 0.045) on seedling fresh weight but its effect was non-significant (p = 0.171) for seedling dry weight. However, genotypic effect was significant for both traits (p<0.01), with the genotype WH 1124 having the highest seedling fresh and dry weight followed by KRL 213 and WH 1105 (Table 5). In terms of trait specific trend, the quantity of variation between genotypes was greater in case of fresh weight than dry weight. Among different soaking durations, maximum seedling fresh and dry weight were reported at 12 hours of hydropriming, which was statistically equivalent to 8 hours. Hydropriming for more than 12 hours, on the other hand, indicated a diminishing tendency. The lowest value of seedling fresh weight was reported in unprimed (dry) seed while seedling dry weight was lowest in 20 hours treatment with a

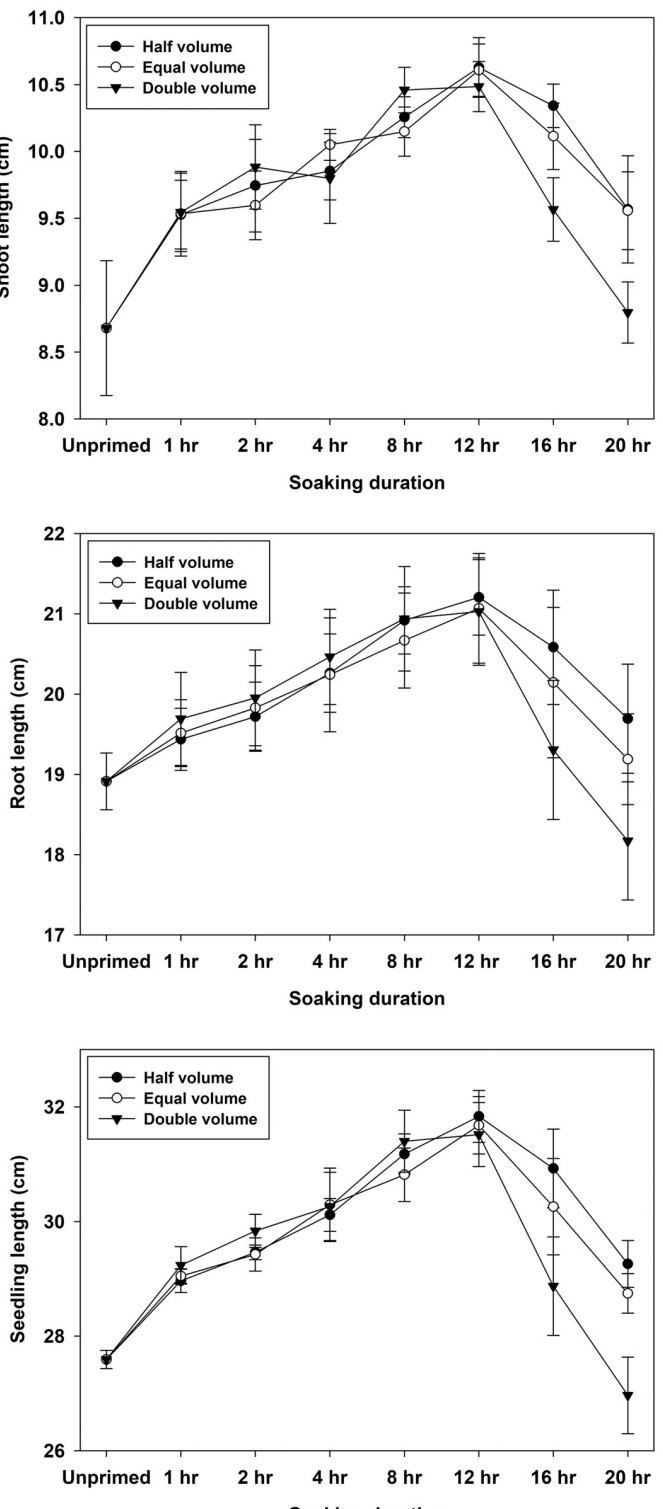

**Fig 4. Interactive effects of water volume and soaking duration of hydropriming on seedling growth characteristics in wheat.**

**Table 5. Main effects of genotype, water volume, soaking duration and temperature on seedling biomass in hyroprimed wheat seed.**

| Main effects | Levels | Seedling fresh weight (mg) | Seedling dry weight (mg) |
|---|---|---|---|
| Genotype, gp | WH 1105 | 133.67 c | 13.60 c |
| | WH 1124 | 185.03 a | 16.15 a |
| | KRL 213 | 140.11 b | 14.02 b |
| Water volume, wv | Half | 153.33 ab | 14.54 a |
| | Equal | 154.28 a | 14.67 a |
| | Double | 151.20 b | 14.56 a |
| Soaking duration, sd | 0 h (Control) | 130.50 e | 13.87 d |
| | 1 h | 141.61 d | 14.37 c |
| | 2 h | 148.46 c | 14.65 bc |
| | 4 h | 157.72 b | 14.89 b |
| | 8 h | 170.19 a | 15.40 a |
| | 12 h | 173.80 a | 15.47 a |
| | 16 h | 158.02 b | 14.46 c |
| | 20 h | 143.19 cd | 13.58 d |
| Temperature, Tm | 20°C | 144.67 b | 13.50 b |
| | 25°C | 161.20 a | 15.67 a |
| LSD$_{gp}$ | | 3.003 | 0.177 |
| LSD$_{wv}$ | | 3.003 | NS |
| LSD$_{sd}$ | | 6.355 | 0.375 |
| LSD$_{Tm}$ | | 2.048 | 0.121 |
| d.f. | | 288 | 288 |

Values with different letters within a column (for each main effect) differ significantly from each other (P < 0.05).

LSD, least significant differences between the treatments; d.f., degrees of freedom for the residual term; NS, Non-significant

statistically similar value at unprimed treatment. A substantial relationship between volume and duration of soaking was also discovered. This interaction revealed that the highest values of seedling fresh and dry weight for half volume and equal volume were observed at 12 hours, whereas hydropriming with double volume produced the highest biomass at 8 hours (Fig 5). Temperature treatments also influenced the biomass accumulation in seedlings and

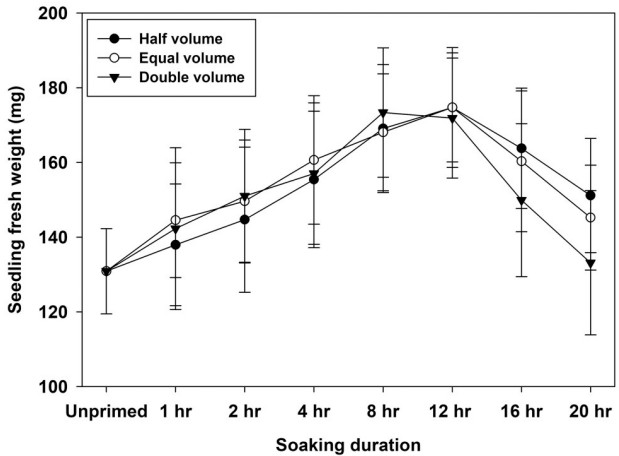 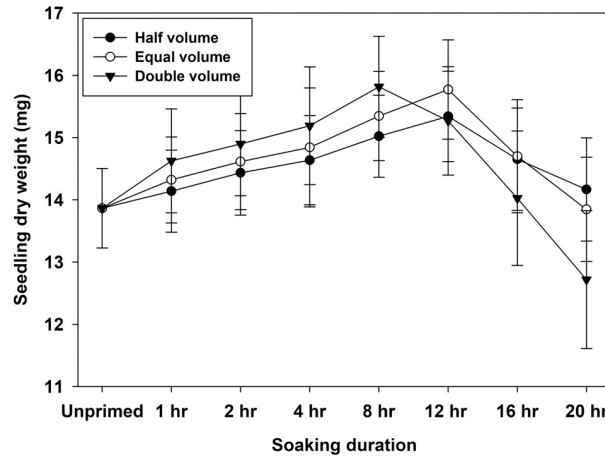

**Fig 5. Interactive effects of water volume and soaking duration of hydropriming on seedling biomass in wheat.**

significantly (p<0.01) lower seedling fresh and dry weights were recorded at 20°C as compared to 25°C (Table 5).

The first order interaction between genotype and water volume indicated that seedling fresh weight of KRL 213 and WH 1124 had no significant differences among the three volume levels. While, the fresh weight of WH 1105, in half and equal volume showed significantly lower value. Similarly, seedling dry weight of KRL 213 had no significant difference for different volume treatments. While, WH 1105 showed lower dry weight at double volume and WH 1124 had significantly lower value at half volume in comparison to other two volume treatments (S1 Table). A significant interaction of genotype and soaking duration was also observed for seedling fresh (p<0.01) and dry (p = 0.014) weights. The genotype WH 1124 outperformed other two genotypes at all the levels of soaking duration for seedling fresh and dry weight. While, WH 1105 had lower values of both these traits at all durations except 8 and 12 hours where KRL 213 was lowest among the three genotypes (S2 Table). First order interaction between genotype and temperature indicated that seedling biomass was maximum in WH 1124 at both the temperatures. WH 1105 was on second rank at 20°C but it performed poorly among three genotypes at 25°C in terms of seedling fresh and dry weights (S3 Table). Similarly, first order interaction between soaking duration and temperature indicated higher vigour at 12 hours duration at both temperature which was statistically equivalent to 8 hours duration. While, lowest vigour was recorded in 20 hours priming at both temperatures. However, the magnitude of difference between durations of priming was higher at 25°C (S7 Table).

**Seedling vigour indices.** The effect of different volume treatments on the vigour index-I was found to be significant (p<0.01). The highest vigour index-I was obtained after hydropriming with half the volume of water, which was statistically equivalent to equal volume followed by double volume (Table 6). In case of vigour index-II, however, the effect was non-significant (p = 0.494). Among the genotypes, the genotype WH 1124 had the highest vigour index-I value, followed by WH 1105 and KRL 213 (Table 5). For vigour index-II, the genotype WH 1124 had the highest value, whereas the genotypes WH 1105 and KRL 213 were recorded with lower values but statistically at par with each other. Seedling vigour indices were significantly (p<0.01) influenced by the temperature treatments. This main effect revealed that the values of both the vigour indices were higher under 25°C than 20°C (Table 6). In terms of priming duration, 12 hours was determined to be the most effective numerically but it was statistically equal to the value found at 8 hours treatment (Table 6). However, 20-hour duration yielded the lowest values of vigour indices, which were significantly lower than the control (unprimed seed). The interaction effect of volume and duration showed that seeds primed for 12 hours using equal volume had the highest vigour index-I followed by 8 hours priming with half volume of water. For vigour index-II, highest value was for 8 hours and double volume which was closely followed by 12 hours and equal volume (Fig 6).

A significant (p<0.01) interaction of genotype and soaking duration was observed for vigour indices. The genotype WH 1124 outperformed other two genotypes at all the levels of soaking duration for both vigour indices. However, genotypes WH 1105 and KRL 213 were statistically at par with each other at all the soaking durations for both vigour indices (S2 Table). The first order interaction between genotype and water volume was significant (p = 0.012) only for vigour index-II. This interaction indicated that vigour of KRL 213 and WH 1124 had no significant differences due to different volume treatments. While, for WH 1105, significant differences were observed at three volume treatments (S3 Table). Interaction between genotype and temperature indicated that vigour indices were maximum in WH 1124 at both the temperatures. However, WH 1105 behaved differently at different temperatures and recorded with higher vigour than KRL 213 at 20°C but lower at 25°C showing its sensitivity for elevated temperature level (S4 Table).

**Table 6. Main effects of genotype, water volume, soaking duration and temperature on seedling vigour indices of hydroprimed wheat seed.**

| Main effects | Levels | Seedling vigour index-I | Seedling vigour index-II |
|---|---|---|---|
| Genotype, gp | WH 1105 | 2763 b | 1284 b |
| | WH 1124 | 2843 a | 1513 a |
| | KRL 213 | 2716 c | 1290 b |
| Water volume, wv | Half | 2801 a | 1361 a |
| | Equal | 2773 ab | 1367 a |
| | Double | 2749 b | 1358 a |
| Soaking duration, sd | 0 h (Control) | 2561 e | 1288 d |
| | 1 h | 2742 d | 1354 c |
| | 2 h | 2817 c | 1395 b |
| | 4 h | 2896 b | 1426 b |
| | 8 h | 3024 a | 1496 a |
| | 12 h | 3087 a | 1507 a |
| | 16 h | 2783 cd | 1340 c |
| | 20 h | 2313 f | 1109 e |
| Temperature, Tm | 20°C | 2555 b | 1263 b |
| | 25°C | 2991 a | 1461 a |
| LSD$_{gp}$ | | 30.957 | 18.752 |
| LSD$_{wv}$ | | 30.957 | NS |
| LSD$_{sd}$ | | 65.518 | 39.687 |
| LSD$_{Tm}$ | | 21.117 | 12.791 |
| d.f. | | 288 | 288 |

Values with different letters within a column (for each main effect) differ significantly from each other (P < 0.05).

LSD, least significant differences between the treatments; d.f., degrees of freedom for the residual term; NS, Non-significant

The second order interaction among genotype, soaking duration and temperature was significant (p<0.01) for vigour index-I (Table 2). It was observed that the genotype WH 1124 outperformed other two genotypes at all the levels of soaking duration at both temperature levels. The genotypes WH 1105 had superior vigour than KRL 213 at all durations at 20°C. But at 25°C temperature, KRL 213 outperformed WH 1105 at all the soaking durations (S5 Table).

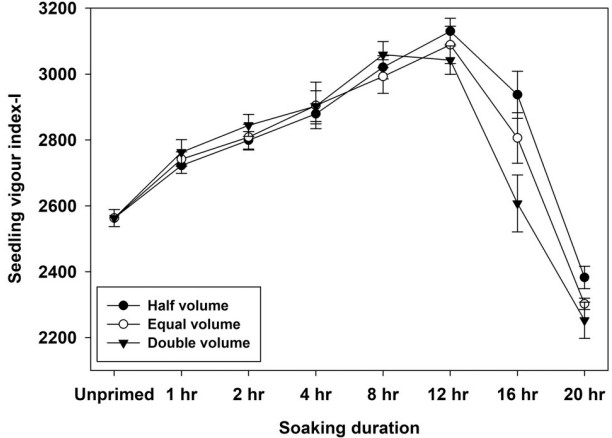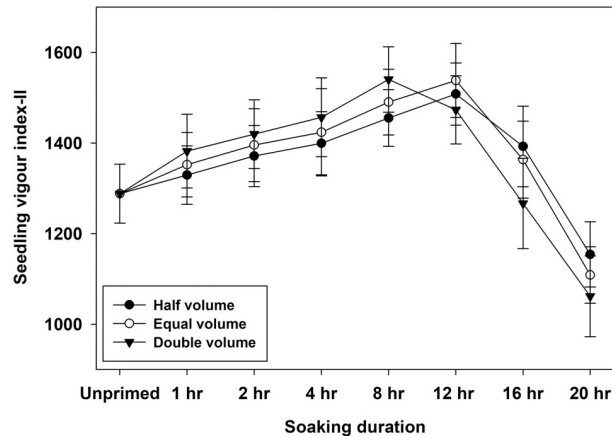

**Fig 6. Interactive effects of water volume and soaking duration of hydropriming on seedling vigour indices in wheat.**

**Table 7. Assessment of different treatment combinations on the basis of Hydropriming Optimization Score (HPOS) in primed wheat seed.**

| Sr. No. | Priming treatments[#] | Water Exposure Index (WEI)[*] | Standard germination (%) (SG) | Germination speed (GS) | HPOS | Ranking |
|---|---|---|---|---|---|---|
| 1. | HV+1 h | 42.43 | 94.00 | 50.97 | 66.10 | 20 |
| 2. | HV+2 h | 84.85 | 95.00 | 57.12 | 71.34 | 15 |
| 3. | HV+4 h | 169.71 | 95.62 | 63.70 | 76.46 | 11 |
| 4. | HV+8 h | 339.41 | 96.89 | 69.95 | 81.24 | 9 |
| 5. | HV+12 h | 509.12 | 98.33 | 75.87 | 85.65 | 3 |
| 6. | HV+16 h | 678.82 | 95.00 | 75.14 | 83.91 | 6 |
| 7. | HV+20 h | 848.53 | 81.45 | 60.89 | 69.68 | 17 |
| 8. | EV+1 h | 60.00 | 94.39 | 55.06 | 69.55 | 18 |
| 9. | EV+2 h | 120.00 | 95.45 | 62.59 | 75.60 | 13 |
| 10. | EV+4 h | 240.00 | 95.89 | 71.03 | 81.61 | 8 |
| 11. | EV+8 h | 480.00 | 97.11 | 75.17 | 84.74 | 5 |
| 12. | EV+12 h | 720.00 | 97.50 | 81.06 | 88.52 | 1 |
| 13. | EV+16 h | 960.00 | 92.78 | 68.76 | 78.98 | 10 |
| 14. | EV+20 h | 1200.00 | 80.11 | 57.92 | 67.23 | 19 |
| 15. | DV+1 h | 84.85 | 94.50 | 56.36 | 70.61 | 16 |
| 16. | DV+2 h | 169.71 | 95.33 | 63.70 | 76.37 | 12 |
| 17. | DV+4 h | 339.41 | 95.95 | 72.59 | 82.65 | 7 |
| 18. | DV+8 h | 678.82 | 97.44 | 78.53 | 86.97 | 2 |
| 19. | DV+12 h | 1018.23 | 96.56 | 75.95 | 85.02 | 4 |
| 20. | DV+16 h | 1357.65 | 90.33 | 61.84 | 73.41 | 14 |
| 21. | DV+20 h | 1697.06 | 83.56 | 51.70 | 63.87 | 21 |
| 22. | Control (Unprimed) | - | 92.89 | 44.06 | 59.77 | 22 |

[#]Soaking in HV- Half volume, EV- Equal volume, DV- Double volume (w/v) with respect to weight of seed + Soaking duration in hours. HPOS = (2×SG×GS)/(SG+GS).

[*]WEI is calculated by following formula: $WEI = \sqrt{wv} \times sd$, where wv = water volume and sd = soaking duration in minutes

**Hydropriming Optimization Score (HPOS).** The evaluation of multiple treatment combinations based on HPOS indicated that all priming treatments were superior than the unprimed control (Table 7). The ranking based on the score was not consistent with the corresponding treatment's germination percentage or germination speed. Rankings, on the other hand, confirmed the interaction effect of soaking duration and water volume used during priming. The top three treatments were 12 hours with equal volume, 8 hours with double volume, and 12 hours with half volume. Similarly, control (dry seed), 20 hours with same volume and 1 hour with half volume, occupied the last three ranks, respectively (Table 7).

**Correlation analysis.** Water exposure index (WEI) and seed moisture content were also correlated with germination and seedling vigour indices. WEI and seed moisture both had a negative relationship with germination percentage and seedling vigour, according to the correlation analysis. WEI showed a strong ($p < 0.01$) negative correlation (r = -600) with standard germination (Table 8). In the case of seedling vigour indices, the association was also negative (r = -0.465 and -0.519 for vigour index-I and II, respectively), but with normal significance ($p < 0.05$) (Table 8). The relationship between WEI and germination speed, on the other hand, was not statistically significant ($p > 0.05$), but it was positive in nature (r = 0.053).

Moisture content of seed had a negative relationship with germination percentage (r = -0.317) and vigour indices (r = -0.120 and -0.156 for vigour index-I and II, respectively), but this was not statistically significant ($p > 0.05$). However, the moisture content of the seed had a positive significant ($p < 0.05$) relationship (r = -0.441) with the speed of germination.

**Table 8. Correlation of water exposure index and moisture content with germination and seedling vigour parameters.**

|        | WEI | Mc | SG | GS | VI-I | VI-II |
|--------|-----|-----|-----|-----|------|-------|
| **WEI** | 1.000 | | | | | |
| **Mc** | 0.892** | 1.000 | | | | |
| **SG** | -0.600** | -0.317[NS] | 1.000 | | | |
| **GS** | 0.053[NS] | 0.441* | 0.571** | 1.000 | | |
| **VI-I** | -0.465* | -0.120[NS] | 0.945** | 0.777** | 1.000 | |
| **VI-II** | -0.519* | -0.156[NS] | 0.950** | 0.748** | 0.984** | 1.000 |

**Significant at 1%,

*Significant at 5%,

[NS]: Non-significant, WEI: Water Exposure Index, Mc: Moisture content, SG: Standard germination, GS: Germination speed, VI-I: Seedling vigour index-I, VI-II: Seedling vigour index-II

## Experiment 2

Another experiment was designed to compare the conventional priming (seed was dried to its original moisture content) with on-farm priming (surface drying). The results revealed that the main effect of drying period after priming significantly influenced most of the characters in all the three genotypes and recorded with lower values in case of conventional method of priming (S11–S13 Tables). However, for standard germination, the effect was non-significant for the genotype KRL 213 (p = 0.422) and less significant for WH 1105 (p = 0.017) and WH 1124 (p = 0.039) (S8–S10 Tables). Similarly, the effect was non-significant for root length (p = 0.194) and seedling fresh weight (p = 0.124) in the genotype KRL 213 (S10 Table).

Critical analysis regarding effect of drying at each volume level suggested that the effect was non-significant for standard germination except equal volume in the genotype WH 1105 (Fig 7). Numerically, the values were lower in conventional priming method. Similarly, the germination speed was significantly lower in conventionally primed seeds at each level of volume in all the three genotypes (Fig 7). For shoot length, all the treatment combinations showed significantly higher values in surface dried seed (on-farm priming) except half volume in the

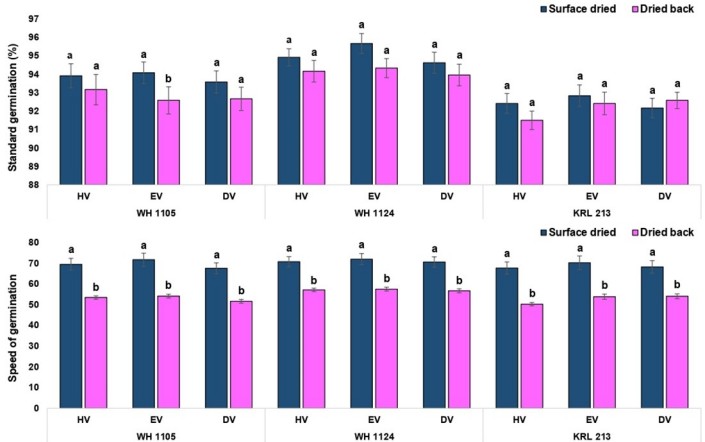

**Fig 7. Germination characteristics as affected by period of drying in case of on-farm priming (surface dried) and conventional priming (dried back to original moisture content) of wheat seed.**

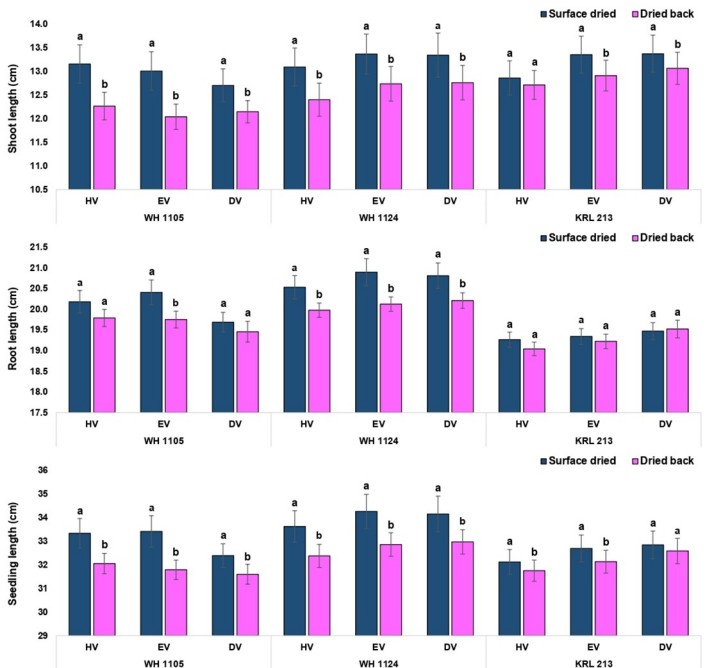

**Fig 8. Comparison between on-farm priming (surface dried) and conventional priming (dried back to original moisture content) with respect to seedling growth parameters of wheat.**

genotype KRL 213. The seeds of the genotype KRL 213 were less affected by the priming method (drying period) as results were non-significant for standard germination, root length and seedling fresh weight at all the volume levels (Figs 8 and 9). In terms of seedling vigour indices, priming methods showed significant differences. In conclusion, on-farm priming is significantly better than conventional method in most of the instances except double volume treatments. Double volume treatment was recorded with at par values of vigour index-II (in KRL 213) and vigour index-II (in WH 1105) for both priming types (Fig 10).

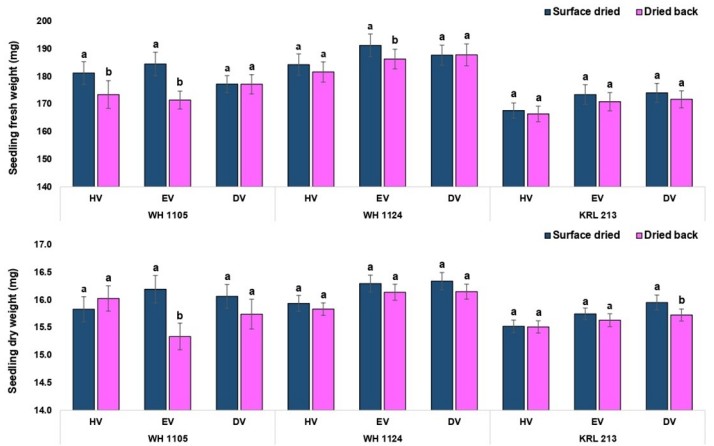

**Fig 9. Comparison between on-farm priming (surface dried) and conventional priming (dried back to original moisture content) with respect to seedling growth parameters of wheat.**

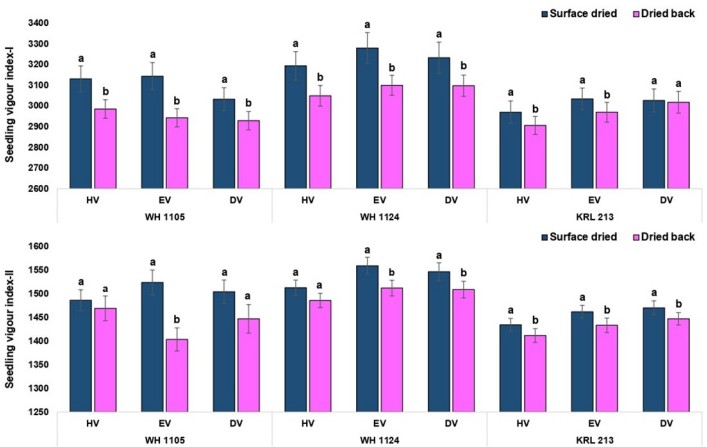

**Fig 10. Seedling vigour as affected by period of drying in case of on-farm priming (surface dried) and conventional priming (dried back to original moisture content).**

## Discussion

Seed priming can help improve seedling emergence and establishment in the field. In a hostile field environment, seed hydropriming's benefits would be highly valuable. Practical aspects are vital because they represent actual application on the farmer's field. The present study attempted to work out optimal duration and water volume for hydropriming by assessing several parameters like water absorption, seed moisture, germination and seedling vigour of three prominent wheat genotypes. The pattern of water absorption under different specifications of hydropriming showed that water is absorbed rapidly at shorter durations and rose dramatically up to 16 hours, after which it decreased significantly. Comparable results were reported in a previous study, with maximum water absorption at 16 hours of soaking in wheat seed [20]. Under all three volume treatments, about half of the water was absorbed within 8 hours. This is a feature of the phase I 'imbibition' stage and shows the fast hydration of interior seed tissues [21]. As imbibition is essentially a passive process that serves as a catalyst for the resumption of metabolic activity, the priming period must be long enough to ensure that germination processes are sufficiently advanced to allow for pre-germinative effects to take place. However, the timing of these events varies based on cultivar, seed quality, and priming specifications.

The hydropriming up to 12 hours enhanced the germination percentage and other seed vigour metrics when compared to the control. Durations greater than 12 hours, i.e. 16 and 20 hours, resulted in a considerable drop in germination and other seed vigour parameters, particularly at the double volume. Hydropriming of wheat for more than 12 hours has previously been reported to be damaging to seed [22]. The standard germination percentage was found to be highest at 12 hours soaking with half and equal volumes at both 20°C and 25°C, however when seeds were primed in double volume, the highest germination was obtained at 8 hours. These findings are similar to some previous reports which suggested that 12 hours hydropriming is best for wheat seed [22–24]. The increased germination of primed seeds could be attributable to readily available nourishment to primed seeds, allowing them to finish the germination process considerably faster than dry seeds [25, 26]. Seed priming also repairs metabolic damage to the genomic DNA caused by desiccation of seeds, enhances mitochondrial membrane quality, seed energy, and ATP/ADP ratio which in turn improves its performance

over unprimed seeds [27]. However, there are some studies in the literature which recorded maximal germination potential in wheat after 16 and 18 hours priming [20, 28].

Genotypic differences revealed that the genotype WH 1124 had maximum germination potential and other seed vigour parameters at both the temperature levels. It was followed by WH 1105, and finally by KRL 213. Genotypic variations are common in germination and seed-ling vigour related traits [29]. The larger seed size of genotype WH 1124 may explain its higher germination and vigour potential when compared to the other two genotypes. The results also revealed that WH 1124 and KRL 213 performed better at higher temperatures i.e. 25°C. These two genotypes were developed specifically for stressed conditions. KRL 213 is a salt tolerant genotype, and WH 1124 is recommended for late sowing. Their better performance at higher temperature suggests that the characteristics that makes a genotype tolerant to one type of abiotic stress can help it perform better under other types of abiotic stresses as well. WH 1105, on the other hand, which is recommended for planting under normal conditions, demonstrated heat stress sensitivity at the germination stage even after priming.

Among the three volume treatments, half and equal volumes demonstrated higher germination and vigour than double volume, where most of the parameters declined. This could be due to a lack of appropriate aeration and development of anaerobic conditions in the seeds dipped in double volume. The rate of water absorption rises as anaerobic conditions develop, potentially causing seed damage [30]. For example, hydropriming with water in the range of 90–100 percent of seed weight was shown to be optimum in lucerne seeds and higher amount of water resulted in reduction of seed performance [19]. While, in case of cowpea [31] and vegetable pea [32] seeds, a volume twice the seed weight produced better results than a half and equal volume, however, the priming duration was shorter (2 hours). These findings suggest that water absorption varies by crop species and should be standardized under diverse set of priming conditions.

The decline of seed performance after a specific priming duration could be attributed to seed degradation caused by unregulated water absorption. Seed/seedling performance reductions in vigour tests at and after a particular timing are unambiguous indications of extremely lengthy priming duration (over-priming). Another factor may be the accumulation of fermentation products in excess as a result of the extended hypoxic conditions during 'on-farm' seed priming conditions, responsible for a gradual loss of vigour [33]. The duration which produces best outcomes of seed priming differs in different crop species/seed type. For example, 18 hours was found optimum for maize [28], for rice 48 hours was found optimum [34], in lucerne it was 3–5 days [19] and for cowpea and vegetable pea, only 2 hours priming was found enough [31, 32].

The study established the interaction effects of volume and length of soaking duration, implying that the extent of water exposure plays a key role in outcomes of hydropriming. Using an ideal water exposure environment throughout the priming process, the germination potential in terms of speed and overall percentage, as well as the growth of subsequent seedlings, can be optimized. The outcomes of 'on-farm' seed priming under varied climatic conditions are mostly influenced by two major criteria for quick germination: (1) the level of hydration of the seed; (2) the advantages of developmental advancement over dry seeds at the time of sowing. Rapid hydration of interior tissues is the primary cause of faster germination in primed seeds [21]. In the present study, the first 4 hours of soaking resulted in a significant improvement in germination speed compared to dry seed (57 percent out of total 76 percent improvement) (Table 2). These findings are consistent with a study in which length of on-farm priming was standardized in barley seeds [21].

The imbibition data show that the seeds had imbibed around 65% of water with respect to their weight at 12 hours priming (Fig 1). As the seeds were not dried back to their original

moisture content and planted just after a brief drying. The advantage here is that the seed do not have to absorb more water from the surrounding substrata immediately. So, the process of germination process began earlier in primed seeds, resulting in higher germination speed as compared to dry seed (Table 2). In this study, it was determined that the rapid boost in germination speed after a few hours of priming is mostly related to the first rapid hydration of seed tissues, whereas developmental advancement is the primary source of improvements at longer soaking times.

Under varying environmental conditions, seedling vigour is the most significant seed quality attribute because it is critical in the establishment of freshly emerged seedlings under hostile field environment [35]. Priming has a significant positive impact on seedling vitality. However, it is highly dependent on the precision and specificity of ideal priming techniques. The highest gain in vigour can be obtained by stopping priming right before the start of phase III. The completion of numerous pre-germinative phenomena, including protein synthesis, mitochondrial synthesis, and other cellular repair mechanisms, is critical in increasing vigour during this stage. However, if the seed enters phase III during the priming process, it will begin cell division and elongation, resulting in radicle protrusion and a loss of vigour obtained by priming [36–38]. Therefore, the optimal priming length should correspond to a stage of maximum molecular production and cellular repair [37, 39, 40]. More practical mechanism behind success of primed seeds is the early germination which results in early establishment of roots and shoots providing developmental advances to growing seedlings. These advantages can act as a strict selective filter in a certain field environment, selecting which seed will establish more quickly [41]. Rapid emergence of seeds in the field leads to better crop establishment, which is conducive to higher yields [42]. Improvement in seedling establishment provides tolerance for biotic (weeds) as well as abiotic (drought, salinity) stresses at initial crop growth stages. Better establishment can also be correlated with higher vigour of individual plants, more tillering capacity, increase in number of fruits and seeds produced per plant, higher seed quality attributes of progeny seeds [43].

In experiment 2, surface drying (on farm priming) for 1 hour was found better than conventional method (redrying to original moisture content) for germination speed and seedling vigour, suggesting the early onset of germination process in the primed seeds (on farm priming). Briefly dried seeds had the advantage in terms of moisture content and they do not have to absorb more water from the surrounding substrata immediately. These results have also been reported in maize [44]. Early seed emergence is advantageous in farming because it improves the plant growth and yield besides protecting plants from various biotic and abiotic stresses [41]. Results also revealed that germination percentage was not significantly influenced by drying period which means that both priming methods are somewhat similar in terms of total germination. But the difference in seedling vigour was significant suggesting the superiority of brief drying (on-farm priming) under variable field environments. However, it was also observed that both the methods of priming showed higher values of germination, seedling growth and vigour as compared to unprimed seeds.

On the resource conservation side, the method of on-farm priming is more pragmatic as it requires less time and other resource inputs. Under rainfed farming conditions, wheat sowing can be done after priming for getting early emergence and better plant stand. The method would be beneficial under irrigated conditions also, especially in paddy-wheat rotation, where time is a crucial resource to avoid late planting and crop establishment which may be good escape from terminal heat stress in wheat crop. Therefore, on the basis of experiment 2, conclusion can be drawn that on-farm priming method is superior in terms of early emergence, initial seedling vigour, time savings, and low input requirement.

## Significance at farmers' field

In agriculture, the true worth of a technique is found in its applicability on the farmer's field. Farmers have known about seed priming for a long time and practice it as per their convenience. They typically soak the seed overnight without taking into account important aspects influencing the process. The findings of present study may be useful in terms of adjusting length and volume based on the actual farm conditions. The current study demonstrates that 12 hours of hydropriming with an equal volume of water to the seed weight works best for wheat seeds.

However, the differences in performance between seeds primed for 8 hours and 12 hours were minor. Furthermore, for durations longer than 12 hours, most traits dropped drastically. As previously stated, the 'safe limits' in the process of seed priming are extremely important, and overpriming can have detrimental impacts on seeds. Therefore, using a 12-hour priming duration on a farm could be risky due to the possibility of errors in both the quantity of water and duration of soaking. Wheat is a winter crop, and the usual sowing season is November, when the nights are longer, so soaking the seeds overnight increases the risk of seed overpriming by farmers. So, it is suggested that a duration of 8–10 hours with double volume of water would be safer and more practical for resource-constrained farmers to implement. Furthermore, these optimal conditions showed a consistent pattern in all three genotypes studied, indicating that they can be applied to other wheat seeds in general.

## Supporting information

**S1 Table. Interactive effect of genotype and water volume of hydropriming on root length, seedling length, seedling biomass and seedling vigour index-I of wheat.**
(DOCX)

**S2 Table. Interactive effect of genotype and soaking duration of hydropriming on germination characteristics, seedling growth and seedling vigour indices of wheat.**
(DOCX)

**S3 Table. Interactive effect of genotype and temperature on seedling growth parameters of wheat.**
(DOCX)

**S4 Table. Interactive effect of genotype and temperature on seedling vigour indices of wheat.**
(DOCX)

**S5 Table. Interactive effect of soaking duration and temperature on shoot length, root length, seedling length and seedling vigour index-I of wheat.**
(DOCX)

**S6 Table. Interactive effect of genotype, soaking duration and temperature on shoot length, root length, seedling length and seedling vigour index-I of wheat.**
(DOCX)

**S7 Table. Interactive effect of genotype, water volume and temperature on root length of wheat seedlings.**
(DOCX)

**S8 Table. Analysis of Variance (F-value) for effects of drying, temperature, water volume and soaking duration on germination characteristics, seedling growth and vigour indices of the wheat genotype WH 1105.**
(DOCX)

**S9 Table. Analysis of Variance (F-value) for effects of drying, temperature, water volume and soaking duration on germination characteristics, seedling growth and vigour indices of the wheat genotype WH 1124.**
(DOCX)

**S10 Table. Analysis of Variance (F-value) for effects of drying, temperature, water volume and soaking duration on germination characteristics, seedling growth and vigour indices of the wheat genotype KRL 213.**
(DOCX)

**S11 Table. Comparison between conventional and on-farm seed priming in case of the genotype WH 1105.**
(DOCX)

**S12 Table. Comparison between conventional and on-farm seed priming in case of the genotype WH 1124.**
(DOCX)

**S13 Table. Comparison between conventional and on-farm seed priming in case of the genotype KRL 213.**
(DOCX)

## Author Contributions

**Conceptualization:** Hemender Tanwar, Virender Singh Mor.

**Data curation:** Hemender Tanwar, Sushma Sharma.

**Formal analysis:** Mujahid Khan, Jitender Yadav, Sonali Sangwan.

**Investigation:** Hemender Tanwar.

**Methodology:** Hemender Tanwar, Virender Singh Mor.

**Project administration:** Virender Singh Mor.

**Resources:** Virender Singh Mor.

**Software:** Mujahid Khan.

**Supervision:** Virender Singh Mor, Axay Bhuker, Vikram Singh, Shikha Yashveer.

**Writing – original draft:** Hemender Tanwar, Sushma Sharma.

**Writing – review & editing:** Virender Singh Mor, Axay Bhuker, Vikram Singh, Sonali Sangwan, Jogender Singh, Shikha Yashveer, Kuldeep Singh.

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
