## [Decision Letter · Decision Letter 0]

31 Aug 2022

PONE-D-22-18783Optimization of ‘on farm’ hydropriming conditions in wheat: Soaking time and water volume have interactive effects on seed performancePLOS ONE

Dear Dr. Tanwar,

Thank you for submitting your manuscript to PLOS ONE. After careful consideration, we feel that it has merit but does not fully meet PLOS ONE’s publication criteria as it currently stands. Therefore, we invite you to submit a revised version of the manuscript that addresses the points raised during the review process.

We look forward to receiving your revised manuscript.

Kind regards,

Aimin Zhang, Ph.D.

Academic Editor

PLOS ONE

Journal Requirements:

Reviewers' comments:

Reviewer's Responses to Questions

**Comments to the Author**

1. Is the manuscript technically sound, and do the data support the conclusions?

Reviewer #1: Partly

2. Has the statistical analysis been performed appropriately and rigorously? 

Reviewer #1: Yes

3. Have the authors made all data underlying the findings in their manuscript fully available?

Reviewer #1: Yes

4. Is the manuscript presented in an intelligible fashion and written in standard English?

Reviewer #1: Yes

5. Review Comments to the Author

Reviewer #1: The data analysis and writing of this paper are relatively good, and the analysis is relatively sufficient. In the part of materials and methods, I have a different opinion. Priming requires water absorption and dehydration. The dehydration process is also very important for priming. It can provide seeds with the ability to cope with water stress and other adversity. The author shortened the drying time to 1 hour. For the field sowing amount, one hour is not enough to dry seeds and is not convenicence to sowing. I don't know why the auther choose to dry for one hour after priming, not like other priming ， drying to the initial moisture content.The paper has not designed a comparative experiment between conventional priming drying and 1-hour drying, which can not prove that 1-hour drying is more effective than traditional priming. If the author supplements the experiment and the real data prove that his design is better than the traditional priming, I think the article can be published.

6. PLOS authors have the option to publish the peer review history of their article (what does this mean?). If published, this will include your full peer review and any attached files.

Reviewer #1: **Yes: **Chunxiangliu

---

## [Author Response · Author response to Decision Letter 0]

3 Dec 2022

Authors are highly thankful to the learned reviewer for the comments. 

The reviewer has raised an important point. 

As per the suggestion, we have conducted an experiment to compare the priming method described in the original manuscript (on-farm priming) and conventional priming (seeds dried to their original moisture content).

The relevant changes have been made in each section of the manuscript. In results section data is presented with help of bar graphs (Fig. 7-10). All the data generated in the experiment have been included in the supplementary section (Supp. Tables 8-13).

The results of the experiment revealed that on farm priming is better in terms of rapid germination and seedling growth and at par in case of germination percentage as compared to conventional priming.

---

## [Editor Report · Decision Letter 1]

12 Jan 2023

Optimization of ‘on farm’ hydropriming conditions in wheat: Soaking time and water volume have interactive effects on seed performance

PONE-D-22-18783R1

Dear Dr. Tanwar,

We’re pleased to inform you that your manuscript has been judged scientifically suitable for publication and will be formally accepted for publication once it meets all outstanding technical requirements.

Kind regards,

Aimin Zhang, Ph.D.

Academic Editor

PLOS ONE
---

## [Editor Report · Acceptance letter]

23 Jan 2023

PONE-D-22-18783R1 

Optimization of ‘on farm’ hydropriming conditions in wheat: Soaking time and water volume have interactive effects on seed performance 

Dear Dr. Tanwar:

I'm pleased to inform you that your manuscript has been deemed suitable for publication in PLOS ONE. Congratulations! Your manuscript is now with our production department. 

Kind regards, 

on behalf of

Prof. Aimin Zhang 

Academic Editor

PLOS ONE